# Construction Information Management: Benefits to the Construction Industry

Peter Adekunle [1,*], Clinton Aigbavboa [1], Opeoluwa Akinradewo [1], Ayodeji Oke [2] and Douglas Aghimien [1]

1   Cidb Centre of Excellence, Faculty of Engineering and the Built Environment, University of Johannesburg, Johannesburg 2006, South Africa
2   Department of Quantity Surveying, School of Environmental Technology, Federal University of Technology, Akure 340271, Nigeria
*   Correspondence: adekunlepeter90@gmail.com

**Abstract:** This paper aimed to unveil the outcome of an appraisal of the systematic approach to construction information management. This was performed with a view of creating awareness of how information management can be beneficial through the management of the large chunk of information emanating from construction processes. This will improve the gathering, sharing, and storage of information for construction activities. The study implemented a quantitative survey approach with the aid of a questionnaire as the mechanism for data gathering from architects, civil engineers, quantity surveyors, mechanical and electrical engineers, construction managers, and project managers. Data analysis ensued with the aid of SPSS in which applicable measure of dispersal and inferential statistics were implemented. The study unveiled that information management is a major aspect of the construction procedure, and that, to achieve in construction, there must be a well-structured information system. Further findings revealed that the benefits of information management include the firm's growth, organizational performance, enhanced market value, employee's motivation, and quality service. The prowess of this study depends on the appraisal of the benefits of systematic management of construction information and how identification of the benefits can help to motivate construction companies.

**Keywords:** Fourth Industrial Revolution; information management; systematic; exploratory factor analysis

## 1. Introduction

Construction projects of high quality require significant and well-planned data processing [1]. Notably, systematic administration of construction information provides a wide range of options for businesses to successfully automate, create, and share their knowledge. Only when knowledge and resources are managed effectively can a specific construction information system be implemented successfully [2]. The explicit and methodical administration of essential knowledge, as well as the related processes of creation, collection, organization, distribution, use, and exploitation, is referred to as knowledge management (KM) [3]. This necessitates transforming individualized knowledge into institutionalized knowledge that can be effectively used throughout an enterprise. In this setting, systematic construction information management is critical to knowledge management (KM). It is argued that KM processes are the core processes for developing a technology's capabilities, and that the efficient use of these processes is becoming increasingly important in the acceptance and deployment of that technology [4]. This emphasizes the critical relevance of information processing (whether computer-based or traditional/manual) and information management in the construction industry [5]. At the same time, it shows that information management has progressed from a minor to a major aspect of institutional management.

Construction is one of the most dynamic, risky, and difficult industries to work in. In addition, the industry has a bad reputation for information management, with many

significant projects missing deadlines and exceeding budgets [6]. Variations in material pricing, labor and plant productivity, and information quality all have a significant impact on this. All too frequently, the risk associated with a poor information management system is either disregarded or dealt with in an arbitrary manner, resulting in an increase in contingency costs on a typical construction project's estimated cost [7]. In a complex industry such as construction, such an approach can lead to significant delays, litigation, and even bankruptcy.

Every day, the potential of utilizing progressive technology through the systematic management of construction information grows [8]. The ability to use new technologies in construction information management (particularly for enhanced productivity, cost reduction, building design, drawing, and project planning in general) is continuously expanding as new software solutions are developed. According to Halou et al. [9] cost management is a necessary requirement for good construction project management. In fact, cost savings are not the only advantage of utilizing advanced technology; it also improves efficiency, reduces the chance of document theft, modernizes the organization, and provides countless other advantages. The appropriate kind of systematic information management for the construction industry would make things easier for the company culture [10]. However, this is not the only benefit associated with systematic information management in the construction industry. Further discussion is provided on these benefits in Section 2. On a single platform, the system saves all construction data. This enables stakeholders to easily access and control any item, including changing its attributes, quantity, or price, as well as checking stock availability. This drastically minimizes the number of errors and physical labor that construction managers must perform.

Other academics believe that it is now vital to examine the other benefits that new technology brings in every field [10–12]. Obtaining more knowledge about the construction industry and its performance can lead to better-informed management decisions, which will have a direct impact on the company's success. Every construction company relies on a variety of systems to receive, distribute, and analyze data in order to fulfill critical objectives [12]. This study aims to fill a gap in the literature by identifying the benefits of construction information management in the digital age. While systematic management of construction information objectives is often overlooked and difficult to market, forward-thinking construction companies reap enormous benefits by aligning specific information management objectives with long-term business objectives. This study details the outcomes of the systematic management of construction information based on this concept. By standardizing the creation, storage, and administration of digital data and information, this research on the benefits of information management in the construction sector can make it simpler for construction teams to work together and guarantee that the information demands of the clients are met.

## 2. Research Gap

Information management underpins productive collaborative connections in the built environment [13,14]. The degree to which the construction team produces accurate information and the extent to which that information is effectively shared with the contractors, builders, and all other stakeholders involved in the project serve as indicators of information management. In order to better adapt to demand changes and conduct business more meritoriously, effective information management enables all stakeholders and parties involved in construction to frequently access the information management system used and obtain real usage statistics [15]. Construction managers rely heavily on the accuracy and dependability of information since the business is characterized by high levels of environmental and demand unpredictability. A manager's response to uncertainty from a contingency viewpoint is to specialize and standardize various activities and duties using information management [16]. Greater insight into inventory problems and assistance in reducing and mitigating uncertainty are both provided by improved information management, whose adoption benefits are yet to be explored. Since information is relevant

to decision-making processes, the impact of information management is increased; the perception of organizational success increases with the quality of the information that is conversed [17]. Construction managers' perceptions of how these agreements will affect the operations of their firms may be influenced by the better information flows made possible by digital technologies. Greater control and administration of the vast amount of information created during construction processes are made possible by improved information flows facilitated by cooperative deployment of digital solutions [18]. The ability to manage construction projects at all levels more precisely thanks to improved control and transparency enhances public views of the advantages of construction information management by lowering uncertainty and transaction costs. The quantification of the benefits that would result from the methodical management of construction information is, therefore, made necessary.

## 3. Construction Information Management

Construction companies must have access to the right data at the right time [19]. This will aid in evaluating the performance of the corporation and subcontractors on a project. Accurate data and analysis help establish the credibility of a possible partner when a systematic management of construction information is implemented [20]. Construction companies work with a variety of other businesses to deliver all of the necessary materials. To analyze the risks connected with each partner, accurate data are required. Predicting outcomes is an important part of data analysis, which is fueled by the systematic management of construction data [21]. It is feasible to plan for diverse outcomes if you can forecast the likely outcomes of a project. Furthermore, through simulations, the firm can learn potential problem areas and, hence, plan for them [22]. These analytics help to recognize the real-world limitations of the project. This will assist in reducing the likelihood of problems arising unexpectedly. Data on the local economy, weather, traffic, and community are required for the project [23]. This information aids in determining the project's phasing. As the first phase is completed, the need for more equipment for the second phase is determined. Analytics also provides construction stakeholders with consistent and easy-to-understand project progress information [24]. Stakeholders will be able to make better decisions with such frequent updates. This also allows workers to track the progress of the construction project. In addition, data analytics will aid in the analysis of the project's impact on parameters such as humidity and temperature. Noting that information management in construction affects all construction-related activities is equally crucial [25]. Construction data are useful information that helps contractors predict the demands of certain construction projects. The benefits that follow from using this data assure a more favorable outcome. The major outcome of construction information systems is that they can review inert, unused data to gain valuable insights. Highlighted below are the benefits of construction information management to the construction industry.

### 3.1. Communication

The significance of construction information communication cannot be emphasized [26]. A breakdown in communication and expectations is the most typical cause of construction problems. Hundreds of individuals are involved in large building projects, with thousands more if manufacturers, salespeople, truck drivers, insurance providers, and other related specialists are included [27]. When implemented, systematic construction information management "provides clarity" to eliminate errors that can occur during the transmission of construction data from one point to another [28]. As construction managers, there are experiences of how a simple lack of communication can quickly escalate. For example, an architect may present a design to a client for approval, after which a long series of events occurs, including detailing, specifying the product and how it will be installed, contractor submittals and shop drawings, ordering the material, traversing the supply and delivery chain, and finally seeing some product on the jobsite. The designated person is then dispatched to install the material or package of parts, demonstrating that the "connection" and

"commitment" have been established. It is important to note that none of these employees were privy to any discussions about the initial plan with the owner or architect [29]. Expectations are defined as a result of the systematic management of construction information. Because of the amount of time it takes from design to supply to delivery, there is great potential for errors, which frequently result in conflicts [30]. Even though the subcontractor or material supplier is clearly liable for replacing the erroneous product due to difficult information transmission, time has been lost, and someone must bear the expense of the error. On the other hand, with systematic management of construction information, these errors are minimized.

### 3.2. Single Point of Contact

Clients benefit significantly from a single point of contact (SPoC) in the construction business's systematic management of construction information, but the construction firm benefits even more with this system in place [31]. To begin, working with an SPoC can ensure that no client information, either from the organization or from the client, is lost [32]. These can be little facts or a complete history of collaboration that could have a favorable impact on the project at hand or future commercial opportunities. Secondly, in a construction environment, employees and stakeholders do not have to look far to find a relevant contact person or gather bits and pieces of information from various departments, saving time and increasing effectiveness because the SPoC can provide consolidated answers and feedback with relevant context [33]. Thirdly, the SPoC is the best person to track client and user happiness over time and suggest adjustments that are targeted to the customer's specific needs, ensuring that clients are satisfied and stay with the business for a long time [34]. Lastly, having a single point of contact for clients helps to establish trust and build good business relationships, which creates fertile ground for promotion or recommendation [35]. It is necessary to provide a single point of contact for clients in order to keep better records, keep them informed, and give better service. Strong records are essential for all types of projects, but construction projects with substantial data are especially critical.

It is critical to understand the client's requests and the changes discussed at each level [36]. With a single point of contact, a single set of communication reviews is required to obtain the entire history of the establishment's interactions with each customer. The client can also keep on track by having a single point of contact [37]. It is easy to identify which communications are the most significant when the client receives all of their communication from one person (which needs urgent attention). In a construction environment, every team member has to know that, if they are in desperate need of information, they can receive it quickly. There is a general improvement in the company's image when a construction firm adopts a systematic methodology for managing information.

### 3.3. Operational Efficiency

Industries that are process-driven, such as oil and gas, pharmaceuticals, and construction, try to improve their operational efficiency [38]. In their operational excellence centers, most companies in these industries have teams and departments dedicated to improving operational efficiency and becoming leaner. Companies can save a large amount of time with the correct construction information management system, potentially resulting in significant value creation, enhanced "competitive advantage", and "increased productivity" for the construction organization and the firm itself [39]. This value can be interpreted as a reduction in the time and effort required to complete the same task. Alternatively, time spent on a task or project might be viewed as cost savings. The bottom line stays dedicated to developing operational excellence through increasing process performance from any standpoint [40]. Without a disciplined strategy, managing construction information can be a stressful and inefficient operation. It is critical to measure the proper things when it comes to expanding or improving operational efficiency [41]. Construction organizations frequently have to devise plans for evaluating various processes, with less emphasis on what counts. Any establishment can gain the most value by improving essential construc-

tion processes. Knowing and understanding what those 'important processes' are, as well as how to assess them, is a crucial first step.

Systematic construction information management can also benefit the project team in achieving the organization's goal of increased operational efficiency [42]. Construction information, such as technical documentation, is often managed by project teams and requires frequent updating. These technical materials are essential for engineers to accomplish their duties safely and successfully. When working with contractors and subcontractors on a project, ensuring that everyone has access to the most recent version is critical. Because many project teams still use email and excel sheets to share crucial engineering information, sharing construction information internally and externally can be a significant burden for project teams [43]. This strategy may put the firm at risk of security breaches. When work orders, such as plant turnarounds and facility inspections, are prepared effectively by ensuring that all of the necessary information is available, significant cost savings can be realized when those projects are completed through the use of a systematic management of construction information. For example, failing to have the necessary information available during a plant turnaround might result in severe delays and cost overruns. It is critical to have accurate information available to all internal and external professionals involved in any building project.

### 3.4. Cost and Schedule

The corporate world is getting increasingly electronically connected [44]. It is critical for the engineering and building industries to be proactive in their response to these seismic shifts. Failure to do so will almost certainly result in a loss of global competitiveness. Because the construction industry is both information-intensive and information-dependent, successful adoption of information technology through systematic management of construction information will surely help to achieve project cost and schedule reduction targets [45]. It is usual practice in the industry to identify specific project-related activities that are crucial to meeting scheduled goals, which is facilitated by rapid decision making [46]. Many of these activities necessitate information exchange. This information exchange method could be internal/departmental or external to the organization. It is difficult to define activity logic at such a micro level that information exchange elements can be detected and tracked [47]. Each time information is transferred, however, project time and resources are depleted.

Improvements in internal processes, external information exchanges, or both result in project cost and schedule reductions through the systematic management of construction data [48]. Information management improvements also improve project quality, resulting in client satisfaction. In actuality, there is great room for process improvement. There are numerous other ideas for enhancing the design and construction process [49]. Information management strategies are just one example of such possibilities. Every new management idea, contract execution strategy, or electronic technology added into the process will certainly have an effect on the total time and cost required to complete all of the tasks common to facility design and construction. These effects, however, must be expected and quantified. Owners, contractors, and other process participants can only speculate without this capability [50]. Given the high expense of implementing these process improvement initiatives, it is critical to go beyond guesswork and use a technology developed to measure process impacts.

### 3.5. Market Insight

Construction enterprises adopting systematic management of construction information are tempted into hiring computer scientists and deploying technical solutions for data processing in the search for insights, thanks to the explosion of big data and quantitative analytics [51]. Companies demand market sensitivity, which data analytics cannot give. While a company's data and knowledge are rapidly expanding, its true insight is not. Because there is no obvious distinction among data, knowledge, and insight, understanding

the concept of insight is important for construction organizations [52]. A market insight, in simple terms, is the discovery of a meaningful, actionable, and previously unappreciated fact about a target market as a result of in-depth, subjective data research. The purpose of insight in marketing, particularly when marketing a previously underused or unknown innovation, is to benefit both parties by satisfying the genuine requirements and wants of the target client while also earning [53]. This means that the greatest market insights are beneficial to all parties involved in construction projects, as well as businesses seeking innovation.

Knowing the value to acquire from market intelligence is just as crucial as comprehending its description [54]. True insights are required for inventions with little to no prior market experience. Once the exact market intelligence required for innovation to thrive is identified, a reliable process for gathering that information may be developed. Once all of the necessary data have been gathered, and construction information has been managed in a systematic manner, there are various different ways to analyze it, depending on which insights are judged the most valuable and essential for the firm's market [55]. Data points such as social media analytics, market growth statistics, and channel sizing can also provide important market information. Market test insights gained through the adoption of a systematic construction information management system can serve as a guidepost for anticipating future needs, forecasting market conditions, revenue generation, appraising competition, and acting quickly in response to newly discovered needs and wants [56]. Rapid market insights are required due to the urgency of technology advances. Today, automated digital market tests can produce worldwide market insights in a fraction of the time and cost, allowing for real-time production of important findings. In the ever-changing, developing, and evolving sector of innovation, rapid insights are required.

From the benefits listed above, it is clear that a thorough literature analysis led to the identification of five constructs. Variables were noted throughout the explanation of the five constructs and are categorized in Table 1. These factors were part of the questionnaire used to assess the benefits of construction information management.

**Table 1.** Extracted benefits from the construct identified.

| Benefits of Construction Information Management | Construct |
| :---: | :---: |
| Provide clarity (BCIM1)<br>Build relationships (BCIM2)<br>Create commitment (BCIM3)<br>Define expectations (BCIM4)<br>Easy transmission of information (BCIM5) | Communication (BCIM1–BCIM5) |
| Efficient service delivery (BCIM6)<br>Accountability (BCIM7)<br>Effective utilization of time (BCIM8)<br>Increased trust and dependability (BCIM9)<br>Improves company's image (BCIM10) | Point of contact (BCIM6–BCIM10) |
| Competitive advantage (BCIM11)<br>Consistency in operational system (BCIM12)<br>Increased productivity (BCIM13)<br>Flexibility of employees (BCIM14)<br>Meeting up with the intended stable cycle time/construction period (BCIM15) | Operational efficiency (BCIM11–BCIM15) |
| Rapid decision making (BCIM16)<br>Collection of quality data (BCIM17)<br>Matching standard requirements (BCIM18)<br>Product performance and variety (BCIM19) | Cost and schedule (BCIM16–BCIM19) |

**Table 1.** *Cont.*

| Benefits of Construction Information Management | Construct |
|---|---|
| Client's satisfaction (BCIM20)<br>Alertness to opportunities (BCIM21)<br>Revenue generation (BCIM22)<br>Improved/better decision making (BCIM23) | Market insight (BCIM20–BCIM23) |

## 4. Materials and Methods

The goal of this study was to assess the basic benefits of comprehensive construction information management. The study adopted the Waffenschmidt et al. [57] standard research methodology, which entails accepting the scope of the investigation and conducting a thorough study of the available literature in relation to the research goals. Following the evaluation, a survey method was used with a questionnaire as the study instrument. The data were analyzed, interpretations were drawn from the findings, and conclusions were drawn. The decision to utilize a questionnaire to collect quantitative data in an investigative approach is based on the fact that the study requested responses from professionals across the construction sector. Using a qualitative strategy such as an interview or other supplementary strategies would have been time-consuming and almost impossible to complete. The questionnaire research was chosen because of its ease of use and ability to reach a wider range of experts in a short period of time [58]. Furthermore, using a questionnaire in an examination can provide computability and objectivity [59]. The questionnaire is one of the most widely used social research methods, which is why it was utilized in this study. The questionnaire used was closed-ended and divided into two sections, the first collecting information on the respondents' identities and the second gathering information on the "outcome" of systematic construction information management. Respondents were given a pentagonal-point Likert scale, with five being strongly agree, four being agree, three being neutral, two being disagree, and one being strongly disagree, to assess 23 categories according to their level of agreement. A five-point Likert scale was used for this study since, theoretically, it is superior to other scales and gives three pieces of information: the direction (positive/negative), the strength of the view, and a neutral point [60]. Utilization often results in an improvement in both response rate and quality. In studies on construction management, risk management in construction, and sustainable construction practices, a similar strategy was used [61–63]. Because of their role in the preparation of construction documents and the supervision of construction projects, the questionnaire was distributed to architects, civil engineers, mechanical and electrical engineers, quantity surveyors, construction managers, construction project managers, and project managers. The questionnaire was only sent to professionals who practiced in South Africa, involved in residential, nonresidential, highways, and every other kind of construction activity. This was achieved in line with Dahabreh et al. [64]; because all construction professionals cannot be examined due to the huge population, only a fraction of the targeted group was assessed. An online questionnaire was chosen for its simplicity of use and diversity of responses. Experts were selected from a large professional internet database and sampled according to their willingness to participate in the review. A snowball method was also advocated in order to extend the data collection duration. This became necessary due to the difficulty encountered in reaching a percentage of these specialists, as certain direct communications were not available. When there is a need to increase the sample size as in the case of Abubakar and Alkassim [65], the snowball approach can be useful. Kirchherr and Charles [66] and Waters [67] also used this methodology. According to the methods advocated, the number of online questionnaires in circulation cannot be predicted, making assessment of an absolute response rate difficult to measure. Nonetheless, 240 responses were collected, which was deemed sufficient for the review based on the argument of Singh and Masuku [68] that more data in the sample increase the authenticity of the information provided. Figure 1 shows the step by step methodology approach for this study.

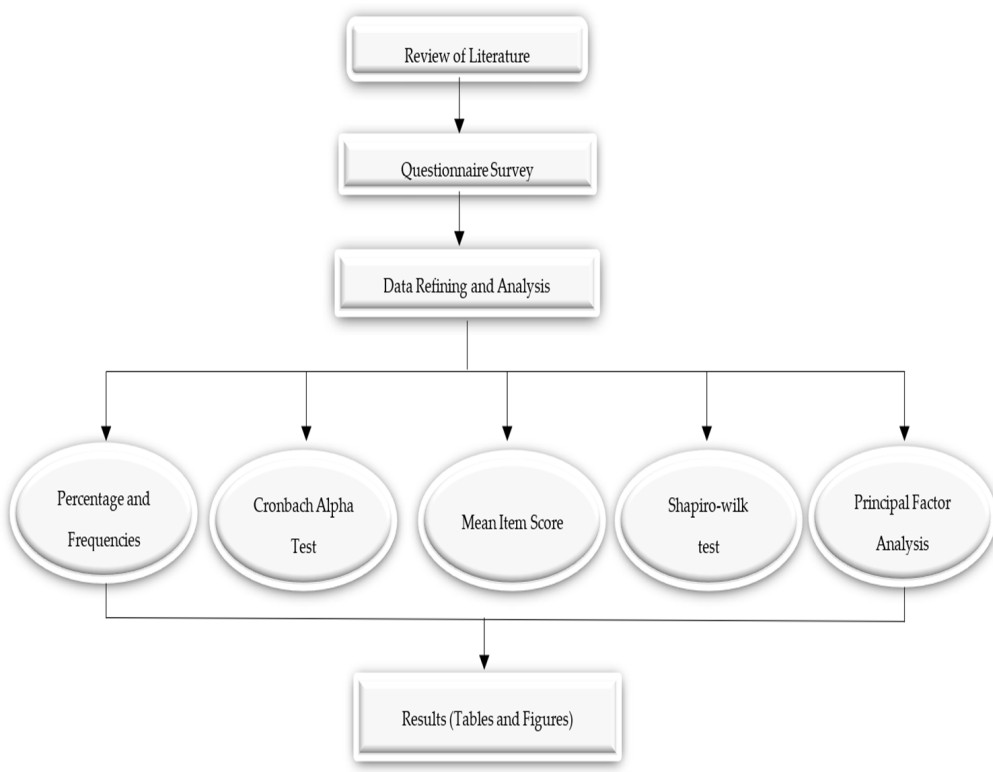

**Figure 1.** Research method adopted.

## 5. Results

The data were examined by first determining their normalcy using the Shapiro–Wilk test and then determining the reliability of the exploration instrument using the Cronbach's alpha test. Cronbach's alpha assigns an alpha value between 0 and 1, with a greater number indicating greater reliability of the data collected [69]. The instrument had an alpha value of 0.890 for the 23 investigated outcomes, indicating its reliability. Kaiser–Meyer–Olkin (KMO) and Bartlett test analyses were used to test the structural validity of the estimated scale. The Bartlett test yielded an estimated chi-square of 1603.23 with 253 degrees of freedom and a significant $p$-value of 0.000, while the outcome yielded a KMO value of 0.725 [70]. The next step was to use the mean item score (MIS) to rank the identified "outcomes" according to their importance degree. This was used in conjunction with the outcomes' inferred standard deviation (SD). These "outcomes" were ordered from most to least important MIS. However, when two factors have identical MIS, Vakili [71] recommends that the variable with the smallest SD be placed first. Furthermore, exploratory factor analysis was used to reduce the large number of variables to a smaller number of summarized variables and to investigate the phenomenon's underlying theoretic structure. The negative factor loadings found in the analysis showed that the formed variables should be viewed in the other direction [72].

The analysis revealed that, in terms of gender, more responses were received from males, which amounted to 68.3% of the responses, while the females accounted for 31.7%. In terms of profession, architects represented 15.0%, civil engineers represented 13.8%, mechanical and electrical engineers represented 20.0%, quantity surveyors represented 24.6%, construction project managers represented 2.5%, construction managers represented 8.3%, and project managers represented 15.8% of the sample. In addition, 45.8% of respondents worked for a contracting firm, 32.5% worked as consultants, and 21.7% worked for the government. Furthermore, over 80% of the respondents had worked on more than six projects while the remaining 20% had worked on six projects and below. On average, more than 80% of respondents had over 5 years of working experience, which is considerably high, making the responses retrieved reliable and credible.

In evaluating the "Benefit of the systematic management of construction information", Table 2 reveals that the most significant outcome was 'improves company's image' with a mean score (M) of 4.96 and standard deviation (SD) of 0.200. Other significant "outcomes" included 'competitive advantage' (M = 4.92; SD = 0.271), 'rapid decision making' (M = 4.90; SD = 0.301), 'client's satisfaction' (M = 4.90; SD = 0.319), and 'meeting up with the intended stable cycle time/construction period' (M = 4.89; 0.366). The table also reveals the results of the Shapiro–Wilk test for normality in which the significant value of all 23 assessed outcomes was well below the 0.05 required criterion for normality. This infers that the information accumulated was nonparametric in nature. The table additionally reveals that every one of the assessed factors gave a mean value higher than the middling value of 3.0, which suggests that respondents accepted every one of the 23 variables as substantial.

**Table 2.** Benefit of systematic management of construction information.

| Benefits of Construction Information Management | $\bar{x}$ | σX | R | Shapiro–Wilk | |
| --- | --- | --- | --- | --- | --- |
| | | | | Statistic | Sig. |
| Improves company's image | 4.96 | 0.200 | 1 | 0.799 | 0.000 |
| Competitive advantage | 4.92 | 0.271 | 2 | 0.898 | 0.000 |
| Rapid decision making | 4.90 | 0.301 | 3 | 0.942 | 0.000 |
| Client's satisfaction | 4.90 | 0.319 | 4 | 0.948 | 0.000 |
| Easy transmission of information | 4.89 | 0.330 | 5 | 0.834 | 0.000 |
| Efficient service delivery | 4.89 | 0.332 | 6 | 0.964 | 0.000 |
| Meeting up with the intended stable cycle time/construction period | 4.89 | 0.366 | 7 | 0.934 | 0.000 |
| Define expectations | 4.88 | 0.327 | 8 | 0.980 | 0.000 |
| Improved/better decision making | 4.85 | 0.358 | 9 | 0.826 | 0.000 |
| Product performance and variety | 4.84 | 0.377 | 10 | 0.841 | 0.000 |
| Provide clarity | 4.83 | 0.381 | 11 | 0.860 | 0.000 |
| Increased trust and dependability | 4.82 | 0.402 | 12 | 0.862 | 0.000 |
| Create commitment | 4.80 | 0.404 | 13 | 0.994 | 0.000 |
| Flexibility of employees | 4.80 | 0.411 | 14 | 0.896 | 0.000 |
| Increased productivity | 4.78 | 0.416 | 15 | 0.911 | 0.000 |
| Revenue generation | 4.77 | 0.462 | 16 | 0.920 | 0.000 |
| Matching standard requirements | 4.74 | 0.450 | 17 | 0.958 | 0.000 |
| Effective utilization of time | 4.73 | 0.462 | 18 | 0.965 | 0.000 |
| Alertness to opportunities | 4.65 | 0.479 | 19 | 0.905 | 0.000 |
| Collection of quality data | 4.62 | 0.494 | 20 | 0.928 | 0.000 |
| Consistency in operational system | 4.57 | 0.496 | 21 | 0.919 | 0.000 |
| Accountability | 4.52 | 0.501 | 22 | 0.936 | 0.000 |
| Build relationships | 4.47 | 0.500 | 23 | 0.935 | 0.000 |

$\bar{x}$ = mean item score; σX = standard deviation; R = rank.

According to Table 3 showing the pattern matrix, the 23 variables recognized from the literature were factored into four clusters inferred on the basis of the perceived inherent connection among the variables in the cluster.

A total of seven variables were loaded onto cluster 1, as shown in Table 3. These variables consisted of 'increased productivity' (83.2%), 'provide clarity' (76.6%), 'revenue

generation' (76.4%) 'competitive advantage' (74.4%), 'efficient service delivery' (71.1%), 'matching standard requirements' (67.4%), and 'easy transmission of information' (47.2%), all of which were related to the growth of the firm. This factor cluster could, therefore, be designated as 'firm's growth' with a variance of 22.795%. It was the highest-ranked factor among the benefits of construction information management.

Six variables were loaded onto cluster 2. These variables consisted of 'product performance and variety' (97.5%), 'increased trust and dependability' (73.6%), 'effective utilization of time' (68.1%), 'client's satisfaction' (65.4%), 'accountability' (58.7%), and 'improved/better decision making' (54.8%). The collective factor of the variables in this cluster was an enhancement in the performance of the firm. The cluster was, therefore, labeled 'organizational performance' with an overall variance of 12.026%. This cluster was ranked second in terms of benefits of construction information management.

Three variables were loaded onto cluster 3, namely, 'collection of quality data' (69.1%), 'alertness to opportunities' (65.4%), and 'consistency in operational system' (56.6%). These variables were related to the positioning of the firm to be marketable. Thus, they were labeled 'enhanced market value'. This cluster gathered 10.073% of the total variance, ranking third in terms of benefits of construction information management.

The fourth cluster comprised two variables: 'build relationships' (90.9%) and 'create commitment' (71.3%). These two factors were related to the benefits received due to the motivation of the employees. Thus, the cluster was labeled '**employee motivation**' with a total variance of 9.490%.

The last cluster comprised five variables: 'improves company's image' (−80.8%), 'rapid decision making' (−55.0%), 'meeting up with the intended cycle time/construction period' (−49.6%), 'flexibility of the employees' (−47.6%), and 'define expectation' (−41.8%). All these factors were related to delivering quality services; thus, the cluster was labeled '**quality service delivery**'. The factor loadings in this cluster consisted of variables with negative values, subsequently indicating their negative correlation with the cluster as previously mentioned in the methodology. This cluster had a total variance of 9.029%, making it the lowest-ranked classification in terms of the benefits of construction information management.

**Table 3.** Factor loading of the benefits of systematic management of construction information.

| Cluster Factor Groupings | Eigenvalues | (% of Variance) | Pattern Matrix Factor | | | | |
|---|---|---|---|---|---|---|---|
| | | | 1 | 2 | 3 | 4 | 5 |
| **Factor 1—Firm's growth** | **3.830** | **22.795** | | | | | |
| Increased productivity | | | 0.832 | | | | |
| Provide clarity | | | 0.766 | | | | |
| Revenue generation | | | 0.764 | | | | |
| Competitive advantage | | | 0.744 | | | | |
| Efficient service delivery | | | 0.711 | | | | |
| Matching standard requirements | | | 0.674 | | | | |
| Easy transmission of information | | | 0.472 | | | | |
| **Factor 2—Organizational performance** | **2.248** | **12.026** | | | | | |
| Product performance and variety | | | | 0.755 | | | |
| Increased trust and dependability | | | | 0.736 | | | |
| Effective utilization of time | | | | 0.681 | | | |
| Client's satisfaction | | | | 0.654 | | | |
| Accountability | | | | 0.587 | | | |
| Improved decision making | | | | 0.548 | | | |

**Table 3.** *Cont.*

| Cluster Factor Groupings | Eigenvalues | (% of Variance) | Pattern Matrix Factor | | | | |
|---|---|---|---|---|---|---|---|
| | | | 1 | 2 | 3 | 4 | 5 |
| **Factor 3 – Enhanced market value** | **1.799** | **10.073** | | | | | |
| Collection of quality data | | | | | 0.691 | | |
| Alertness to opportunities | | | | | 0.654 | | |
| Consistency in operational system | | | | | 0.566 | | |
| **Factor 4—Employee motivation** | **1.365** | **9.490** | | | | | |
| Build relationships | | | | | | 0.909 | |
| Create commitment | | | | | | 0.713 | |
| **Factor 5—Quality service delivery** | **1.259** | **9.029** | | | | | |
| Improves company's image | | | | | | | −0.808 |
| Rapid decision making | | | | | | | −0.550 |
| Meeting up with the intended stable cycle time/construction period | | | | | | | −0.496 |
| Flexibility of employees | | | | | | | −0.476 |
| Define expectations | | | | | | | −0.418 |
| **Total variance explained** | | **63.413** | | | | | |

After using exploratory factor analysis to identify the correlation patterns present in the dataset, the abovementioned five-factor clusters are described below.

Cluster one—firm's growth: Several empirical studies [1,4,5] have explained the importance of information management in the growth of the firm. The motivation for growth or the intention to grow is associated with an efficient and effective "systematic management of construction information". From the moment a firm is being created, it is important to orientate every associate toward growth, effective service delivery, and possessing a competitive advantage in the industry. It is expected that the firm's objective is aligned with the firm's vision and mission.

Cluster two—organizational performance: To successfully experience organizational performance, there must be an understanding of the organizational improvement outcomes. The concept compares the goals and objectives of the organization with its actual performance in the areas of information management [10]. It is proven that organizations with strong information management systems inspire confidence amongst their employees and consumers. Employees look up to well-structured instructions for directions because they have faith in the prowess of valid information [73]. Sharing information empowers the employees with practical knowledge, and this ultimately boosts organizational performance [74]. Notably, when a company is well informed, it nurtures the organization's performance so that it can become among the best in the market.

Cluster three—enhanced market value: Flyvbjerg [75] stated that acquiring esteem from data management is one of the principle worries of construction managers. One of the actions that can be utilized to demonstrate the business worth of data to the management is through training. According to Al-Aomar [76], organizations that effectively integrate an information management strategy with their business plans do so by concentrating on information as a carrier of value and source of competitive advantage. Kaynak [77] stated in a study on the business value of information management that there is debate about the impact of information on organizational performance. The superior opinion is that investment in information technology (IT) through the adoption of a systematic management of construction information can boost the business and economic value of the establishment. Quantifying the business values can then justify the significance of investments on construction information management.

Cluster four—employee motivation: According to Pereira et al. [21], construction establishments have made numerous investments to foster growth toward accomplishing stated targets. However, after developing goal-oriented strategies for each employee and publicizing the firm's vision, it is recognized that systematic construction information management is a motivator for the construction team to achieve greatness. The truth is that simply conveying goals and objectives is insufficient to inspire each team member without a system in place. The company's basis might swiftly crumble if these aims are not communicated continuously (through systematic management of construction information) [78]. It is the obligation of an organized systematic management of construction information to gather individual responsibility from every worker to improve achievement. Positive communication is one of the most effective strategies to do so. Because motivation can take various forms, the company's communication network must elicit an innate urge to cross items off a to-do list and act with intention and purpose toward the organizational goal.

Cluster five—quality service delivery: Most construction establishments make an effort to control quality; however, generally, most of them do not have a robust quality management process in place [79]. In many cases, they do not have a written program either. Conventionally, the project manager is responsible for the quality of the work, and the project coordinator depends on the different craft workers to follow regular and conventional industry practices when it comes to the quality of the work [80]. Such a process depends greatly on the ability, knowledge, discretion, and diligence of workers and the supervisor's persistent and careful oversight with the guide of the predetermined project information (which includes the design, specification, bill of quantities, and other information sources). Many elements become the most significant feature that must be well managed in such a flexible administration framework to ensure that the nature of the activity meets objectives and predictions.

## 6. Implication of Study

Construction businesses of all sizes and locations are creating new and more effective ways to use data to generate commercial plans, while simultaneously considering the ethical implications of processing massive amounts of data related to construction activities. In unison with the global digital infrastructure, the discipline of information management is fast evolving. Construction data are critical to the existence of any construction company. They have an impact on how construction companies formulate strategy and put those ideas into action. They are at the heart of business growth, which is why so much effort and money is spent developing excellent information management systems and enlisting the help of competent professionals to implement them. Figure 2 depicts the model that displays the "outcomes or benefits of the systematic management of construction information" as a function of the cluster components derived. This model also demonstrates that adopting a systematic approach to construction information management leads to enhanced efficiency, higher earnings, and a competitive advantage in the sector. Organizations are increasingly under pressure to manage data responsibly and ethically, as well as to comply with ever-changing legal requirements. Clients and governments are demanding more transparency; therefore, managing information about the organization's operations is key, and the requirement for efficient, safe, and effective management information systems is more important than ever. It is, therefore, fundamental to explore the benefits of systematic management of construction information to enhance the adoption of the systematic management of construction information.

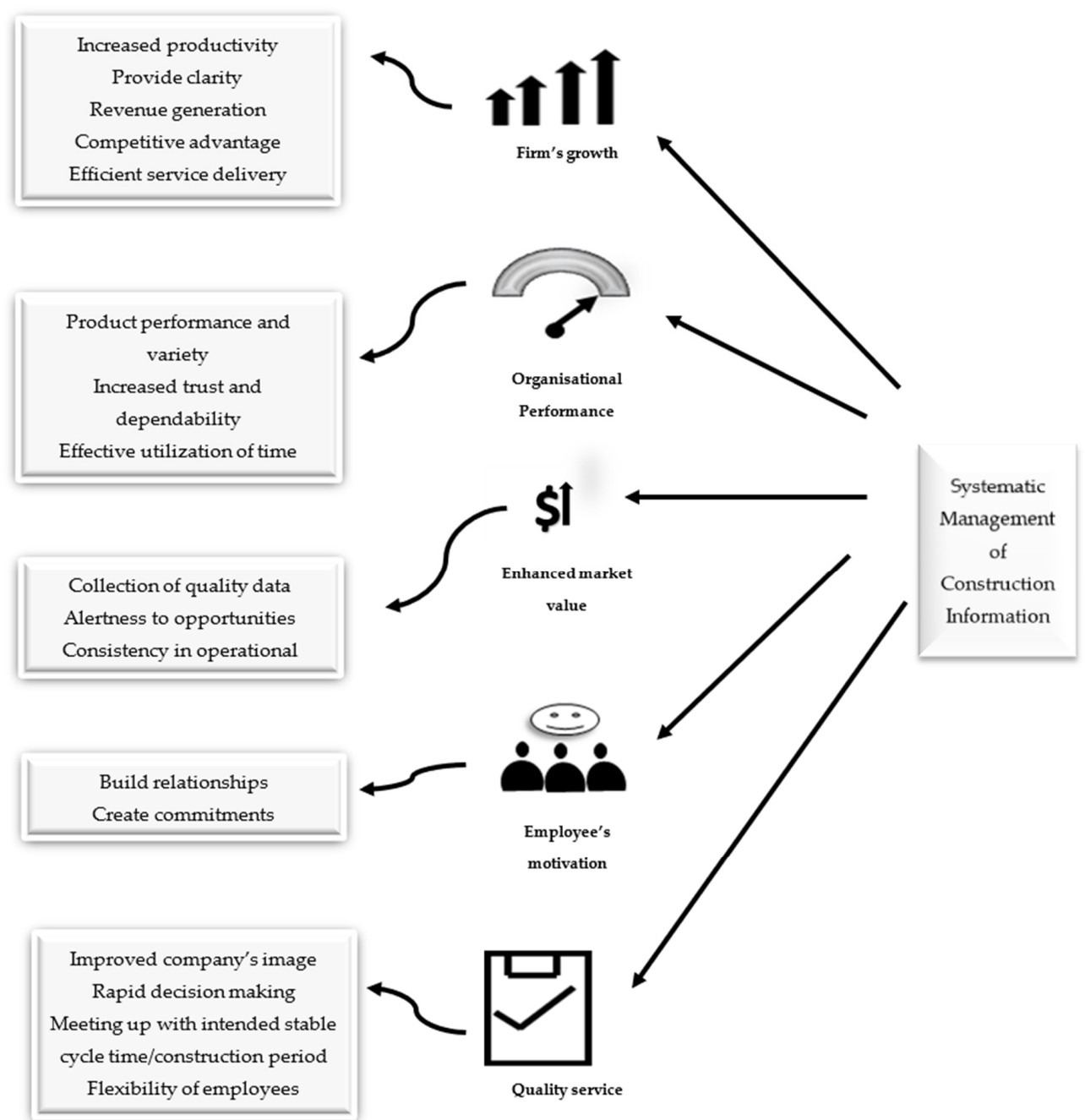

**Figure 2.** Systematic management of construction information outcome model.

## 7. Conclusions

This study set out to assess the benefits of systematic management of construction information for successful execution of construction projects. To achieve this, a quantitative survey was adopted, and primary data were retrieved from professionals in the South African construction industry. Primary data retrieved from respondents revealed that the factors categorized as "firm's growth" were the most beneficial elements of construction information management. Respondents are of the opinion that systematic management of construction information gives room for the provision of clarity of information needed per time on a construction site, helps in improving synergy among construction professionals, and helps to create commitment to the construction project.

The discoveries above uncovered that the opportunities of construction information managements can be measured. The significance of incorporating an effective information

management system has been recognized as a vital part of achieving success and completion of projects in the construction industry. As reported by the respondents, communication amidst the stakeholders involved in a construction project will be optimized through information management. With a proper information management system put in place, construction professionals can have access to the right information at the right time. This study, therefore, recommends that it is important for construction managers to understudy a project to determine the information system that will be suitable for that project during the phase of execution. In addition to ingraining the right sort of work ethics in practice, construction professionals encounter an energized improvement of working conditions, subsequently working on their productivity according to the provision and development of standard information systems.

The study was limited to construction professionals within South Africa and, thus, gives room for further study to be carried out to have an overview of the opinion of professionals in developed countries. On the basis of the results of this study, future studies should seek to investigate the perception of potential clients regarding the benefits accrued from the effective management of construction information. In addition, it will be crucial to assess the challenges of managing construction information.

**Author Contributions:** Conceptualization, P.A., O.A. and C.A.; methodology, P.A. and O.A.; software, P.A. and C.A.; validation, O.A., C.A., A.O. and D.A.; formal analysis, P.A.; investigation, P.A. and O.A.; resources, C.A. and O.A.; data curation, P.A.; writing—original draft preparation, P.A.; writing—review and editing, O.A. and D.A.; supervision, O.A. and C.A. All authors have read and agreed to the published version of the manuscript.

**Funding:** This research was funded by the Center of Excellence, Faculty of Engineering and the Built Environment, University of Johannesburg, South Africa.

**Institutional Review Board Statement:** This study was conducted according to the guidelines of the University of Johannesburg's Research Ethics Committee.

**Informed Consent Statement:** Not applicable.

**Data Availability Statement:** Anonymized data are available from the corresponding author upon written request and subject to review.

**Conflicts of Interest:** The authors declare no conflict of interest.

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
