# Peer review of "Construction Information Management: Benefits to the Construction Industry"

_sustainability, doi:10.3390/su141811366_

Round 1

Reviewer 1 Report (New Reviewer)

I would like to thank the authors for their manuscript. I think it is a well-written document that explores and important topic within the construction management body of knowledge. I have a few minor comments: 

1. The authors should offer more context associated with the South African construction industry and about the background of survey participants. The construction industry may incorporate a myriad of sectors, which may be not directly related: residential, non-residential, highways, transportation, etc. Please clarify. 

2. The authors need to explain how the literature review is related to the questionnaire survey. How did they establish the questions and the outcomes? Currently we do not have a clear relationship between these two sections because section 2 is about topics not discussed in sections 3 and 4. Please show how you determined the themes associated with your questions and outcomes. 

3. In the Conclusion, I believe it is useful to have a brief analysis of how this study could be replicated in other contexts and jurisdictions. 

Author Response

Point 1: The authors should offer more context associated with the South African construction industry and about the background of survey participants. The construction industry may incorporate a myriad of sectors, which may be not directly related: residential, non-residential, highways, transportation, etc. Please clarify.

Response 1: Noting that information management in construction affects all construction-related activities is equally crucial [19]. Construction data is useful information that helps contractors predict the demands of certain projects in every construction project. The benefits that follow from using this data assure a more favorable outcome.

The questionnaire was only sent to professionals who practiced in South Africa and they are professionals involved in residential, non-residential, highways and every other kind of construction activity.

Point 2: The authors need to explain how the literature review is related to the questionnaire survey. How did they establish the questions and the outcomes? Currently we do not have a clear relationship between these two sections because section 2 is about topics not discussed in sections 3 and 4. Please show how you determined the themes associated with your questions and outcomes.

Response 2:

“From the benefits listed above, it is clear that a thorough literature analysis led to the identification of five (5) constructs. Variables were noted throughout the explanation of the five constructs and are categorized in the table 1 below. These factors were part of the questionnaire used to assess the benefits of construction information management.”

Table 1. Extracted benefits from the construct identified

Benefits of Construction information management

Construct

Provide Clarity (BCIM1)

Communication (BCIM1 – BCIM5)

Build relationships (BCIM2)

Create commitment (BCIM3)

Define expectations (BCIM4)

Easy transmission of information (BCIM5)

Efficient service delivery (BCIM6)

Point of Contact (BCIM6 – BCIM10)

Accountability (BCIM7)

Effective utilization of time (BCIM8)

Increased trust and dependability (BCIM9)

Improves company’s image (BCIM10)

Competitive advantage (BCIM11)

Operational efficiency (BCIM11 – BCIM15)

Consistency in operational system (BCIM12)

Increased Productivity (BCIM13)

Flexibility of employees (BCIM14)

Meeting up with the intended stable cycle time/construction period (BCIM15)

Rapid decision making (BCIM16)

Cost and schedule (BCIM16 – BCIM19)

Collection of quality data (BCIM17)

Matching standard requirements (BCIM18)

Product performance and Variety (BCIM19)

Client’s Satisfaction (BCIM20)

Market insight (BCIM20 – BCIM23)

Alertness to opportunities (BCIM21)

Revenue generation (BCIM22)

Improved/better decision making (BCIM23)

Point 3: In the Conclusion, I believe it is useful to have a brief analysis of how this study could be replicated in other contexts and jurisdictions.

The study was limited to construction professionals within South Africa and thus gives room for further study to be carried out to have an overview of the opinion of professionals in developed countries. Based on the result of this study, future study should seek to investigate the perception of potential client regarding the benefits accrued from effective management of construction information. In addition to this, it will be crucial to also access the challenges of managing construction information.

Reviewer 2 Report (New Reviewer)

Dear authors, a few questions, suggestions for your paper:

-what is a "SPSS" software package (line 17)

-no (exemplary) textual or graphic digital support aids (such as 3D collision checks, e.g. BIM Collab) are mentioned in the further course of your text

-How to organize the primary goals of project management:

Costs

Dates

Qualities

in your investigation? Where are you questioned in your Questionaere?

-Does your presentation ONLY refer to possible benefits from the point of view of the (internal) advantages for the construction companies? If so, what would be the reference to overarching sustainability as a substantive "bracket" for the paper?

-Note: Source Waffenschmidt is not numbering 51, but 50 (line 269)

Best regards 

Author Response

Point 1: what is a "SPSS" software package (line 17)

Response 1: It was corrected to “SPSS” alone.

Point 2: no (exemplary) textual or graphic digital support aids (such as 3D collision checks, e.g. BIM Collab) are mentioned in the further course of your text.

Response 2: Construction information management encompasses all 4IR technologies and devices, so that is why a generic name was used to describe the tools/devices.

Point 3: -How to organize the primary goals of project management: Costs, Dates, Qualities in your investigation? Where are you questioned in your Questionaere?

Response 3: As initially ponted out by the Reviewer 1, the identified subsections in section 2 represents the constructs from which the variables to measure the benefits were extracted. The variables have been clearly outlined and tabularized.

Point 4: - Source Waffenschmidt is not numbering 51, but 50 (line 269)

Response 4: This has been corrected

Reviewer 3 Report (New Reviewer)

The following comments can improve the article:

1. The author must highlight the literature gap and novelty of the study in the last paragraph of the Introduction section. 

2. A separate Literature review section must be added to describe the research background and literature gap.

3. In the Methodology section:

A- Why a 5 points Likert scale, why not something else?

B- There exist guidelines for expert elicitation. The sample size is always important in science.

C- The experts' specification is unclear, this should be better highlighted.

D- Authors must show the reliability of the questionnaire, also. 

4. Results are clearly presented, but this manuscript needs an in-depth explanation.

5. The conclusion section must be improved.

Author Response

Point 1: The author must highlight the literature gap and novelty of the study in the last paragraph of the Introduction section.

Response 1: “This study aims to fill a gap in the literature by identifying the benefits of construction information management in the digital age.”

The novelty of the match was highlighted im the first lines of the last paragraph of the introduction session.

Point 2: A separate Literature review section must be added to describe the research background and literature gap.

Response 2: 2. Research Gap

Information management underpins productive collaborative connections in the built environment [13; 14]. The degree to which the construction team produces accu-rate information and the extent to which that information is effectively shared with the contractors, builders and all other stakeholders involved in the project serve as indica-tors of information management. In order to better adapt to demand changes and conduct business more meritoriously, effective information management enables all stakeholders and parties involved in construction to frequently access the information management system used and obtain real usage statistics [15]. Construction managers rely heavily on the accuracy and dependability of information since the business is characterized by high levels of environmental and demand unpredictability. A man-ager's response to uncertainty from a contingency viewpoint is to specialize and standardize various activities and duties using information management [16]. Greater insight into inventory problems and assistance in reducing and mitigating uncertainty are both provided by improved information management, which adoption benefits are yet to be explored. Since information is relevant to decision-making processes, the im-pact of information management is increased; the perception of organizational success increases with the quality of the information that is conversed [17]. Construction man-agers' perceptions of how these agreements will affect the operations of their firms may be influenced by the better information flows made possible by digital technolo-gies. Greater control and administration of the vast amount of information created during construction processes are made possible by improved information flows facil-itated by cooperative deployment of digital solutions [18]. The ability to manage con-struction projects at all levels more precisely thanks to improved control and trans-parency enhances public views of the advantages of construction information man-agement by lowering uncertainty and transaction costs. The necessity for quantifying the benefits that would result from the methodical management of construction in-formation is therefore made necessary.

Point 3: -In the Methodology section:

A- Why a 5 points Likert scale, why not something else?

B- There exist guidelines for expert elicitation. The sample size is always important in science.

C- The experts' specification is unclear, this should be better highlighted.

D- Authors must show the reliability of the questionnaire, also.

Response 3: A- “The use of five-point likert scale was used for this study since, theoretically, it is supe-rior to other scales and give three pieces of information: the direction (posi-tive/negative), the strength of the view, and a neutral point [60]. Utilization often re-sults in an improvement in both response rate and quality.”

B- The experts employed for this study as reported in the research were professionals in the construction industry.

C- “In terms of profession, architects were 15.0%, civil engineers were 13.8%, mechanical and electrical engineers were 20.0%, quantity surveyors were 24.6%, construction pro-ject managers were 2.5%, construction managers were 8.3% and project managers were 15.8%.” This was already included in the report.

D- “The instrument has an alpha value of 0.890 for the 23 investigated outcomes, indicating that it is reliable.” This was already invluded in the study.

Point 4: - Results are clearly presented, but this manuscript needs an in-depth explanation.

Response 4: Further expalnation done as suggested by the other reviewers.

Point 5: - The conclusion section must be improved.

Response 5: . The conclusion was improved on by recommending other context in which this study can be carried out. Overemphasizing on the constrution can derail from the points the authors are trying to make.

Round 2

Reviewer 3 Report (New Reviewer)

The authors answered my comments and no further comments.

This manuscript is a resubmission of an earlier submission. The following is a list of the peer review reports and author responses from that submission.

Round 1

Reviewer 1 Report

While the manuscript is well written and has a scientific methodology for collecting and analyzing data, the topic of the study is well recognized and researched by many researchers over decades. There have been numerous advances and technologies developed to address all these challenges. Unfortunately, there are no new contributions or knowledge that are being presented and/or developed by this study. I am compelled to decline the publication of this manuscript. It may be better suited as an article in a magazine or a newspaper. 

Reviewer 2 Report

The authors present an interesting and a timely work on the need for having effective Construction Information Management systems and their associated benefits. The article builds on the need for such systems based on the extant literature and highlights specific areas on a temporal basis, which can help improve construction workflow processes. However, the paper lacks clarity on a few important aspects which can help strengthen the quality from its current form.

a) Introduction - The authors need to clearly define 'Systematic Construction Information Management' from the extant literature. While the authors describe the need for having such systems set up at a project level, what are the benefits at an organizational level in terms of overcoming the traditional barriers to knowledge management? Is organizational culture the only benefit organizations can reap or are there any ultimate benefits organizations can expect?

b) Construction Information Management - The authors need to specify the basis for selecting the factors mentioned within the subsections. Are these factors all-encompassing with regards to the benefits of Construction Information Management systems? Moreover, is/are there any overlaps between two or more factors mentioned? The authors are advised to perform a thorough literature review to better answer this question.

c) Materials and Methods - The authors are advised to depict the same using a figure that presents a step-by-step methodology adopted and the expected outcomes at the end of each step. Moreover, adding this figure will allow the authors to fine-tune the verbose within this section. This will provide the reader with enhanced content to peruse while maintaining their attention span.

The last paragraph of this section should be moved to the Results section.

This section requires additional information in terms of the total number of surveys sent, and final number of completed surveys received - this will help the readers understand the rate of success of survey responses.

d) Results: The authors specify in the previous section "These "outcomes" were ordered from most to least important MIS. However, when two factors have identical MIS, Vakili recommends that the variable with the smallest SD be placed first" Table 2 has similar MIS values for a few factors. The beauty of continuous variables is that they can always be differentiated based on their numerals obtained. The authors are advised to separately rank all the 23 individual factor, with an extra decimal value for sd, if needed.

e) The remaining sections look alright. However, based on my previous comments, these sections will need appropriate modifications accordingly.

A few additional observations: The style of language needs modifications. For example, some places contain the word 'you' within the text. For example, line 90: "This will assist you in reducing the likelihood of problems arising unexpectedly."

Line 348: There is no Table 7.19.